

# *Primulina anisocymosa* (Gesneriaceae), a new species with a unique inflorescence structure from Guangdong, China

Xin Hong[1,2,3], Jeremy Keene[4], Zhi-Jing Qiu[5] and Fang Wen[2]

[1] School of Resources and Environmental Engineering, Anhui University, Hefei, Anhui, China
[2] The Gesneriad Conservation Center of China, Guangxi Key Laboratory of Plant Conservation and Restoration Ecology in Karst Terrain, Guilin Botanical Garden, Guangxi Institute of Botany, Guangxi Zhuang Autonomous Region and Chinese Academy of Sciences, Guilin, China
[3] College of Life Science, Anhui Normal University, Wuhu, Anhui, China
[4] Glenville State College, Glenville, WV, USA
[5] Laboratory of Southern Subtropical Plant Diversity, Fairylake Botanical Garden, Shenzhen and Chinese Academy of Sciences, Shenzhen, China

## ABSTRACT

A new *Primulina* species from Guangdong, China with an unusual inflorescence is described here. *Primulina anisocymosa* is vegetatively most similar to *P. bobaiensis*. It can be distinguished from all species within *Primulina* morphologically by its unique zigzag monochasial cyme and infructescence. To confirm the phylogenetic relationships and generic placement of this species, not only morphological anatomical features but also chromosome and DNA sequence data were examined and analysed here. Two samples from different populations identified as *Primulina anisocymosa* are monophyletic and were nested in a monophyletic clade within *Primulina* with high branch support. The somatic chromosome number of the new species is also reported ($2n = 36$), supporting its placement in the genus.

## INTRODUCTION

*Primulina* Hance (Didymocarpinae: Gesneriaceae) was originally described as a monotypic genus with *P. tabacum* Hance the only known species (*Wang et al., 1998*; *Li & Wang, 2004*; *Zheng & Xia, 2005*; *Wei et al., 2010*). *Primulina* was recently expanded to include other Asian genera making the genus one of the largest within Gesneriaceae. The genus is distributed from southern China southward into central Vietnam (*Möller et al., 2016*) and comprises more than 174 recognized species. Thus far, only 19 species are reported from Vietnam, the remainder occur in China (*Hô, 2000*; *Weber et al., 2011a*; *Möller et al., 2016*).

Some recent studies have suggested that the vast majority of *Primulina* species grow on limestone in tropical and subtropical Karst areas (*Xu, 1993*; *Ai et al., 2015*; *Kang et al., 2014*). Most of the known species from China and Vietnam share a similar inflorescence structure (dichotomous cyme). This inflorescence is highly variable among species. *Primulina diffusa* Xin Hong, F Wen and SB Zhou for example has a single-flowered

Corresponding author
Fang Wen, wenfang760608@139.com

cyme, which is an extreme reduction of the dichotomous cyme (*Li & Wang, 2004*; *Wang et al., 1998*; *Wei et al., 2010*). The significant variation in inflorescence structure and corolla morphologies suggests a high degree of evolutionary lability, presumably driven by adaptation to different pollinators (*Weber et al., 2011b*). Due to the wide variation in characteristics within the Asian genera it is difficult to ascertain a single synapomorphy for the genus (*Chen et al., 2014*). Thus, it is necessary to have molecular evidence to effectively place unique species in the right genus.

In 2009, we found a population of *Primulina* with only the previous year's fruit in Yangchun city, Guangdong province. Then, in the following year, Mr. Wei-Jun Wu (WJW) found the same purported species on another limestone hill near Gaozhou city, Guangdong province. We observed the species closely and noted that it was an undocumented species of *Primulina*, based on morphology. The species looked quite different from other *Primulina* species based on the infructescence which looked similar to a cincinnus, a determinate cymose inflorescence with a zigzag rachis. In the following year, FW and WJW collected some specimens with flowers from the same locality for study. They observed the development of the flowers and the unusual zigzag monochasial inflorescence and infructescence. To confirm the phylogenetic relationships and generic placement of this species, chromosome and DNA sequence data were collected and analysed. After consulting the relevant literature (*Wang et al., 1998*; *Hộ, 2000*; *Li & Wang, 2004*; *Wei et al., 2010*) along with the chromosome characteristics and molecular analysis, we concluded that this new species was assignable to *Primulina* (*Weber et al., 2011a*). We provide a description and illustration of the new species here.

## MATERIALS AND METHODS

### Ethics statement

All the collecting locations of the new species reported in this study are outside any natural conservation area and no specific permissions were required for these locations. Since the species are currently undescribed, they are not currently included in the China Species Red List (*Wang & Xie, 2004*). Our field studies did not involve any endangered or protected species. No specific permits were required for the present study.

### Nomenclature

The electronic version of this article in Portable Document Format (PDF) will represent a published work according to the International Code of Nomenclature for algae, fungi, and plants (ICN), hence the new names contained in the electronic version are effectively published under that Code from the electronic edition alone. In addition, new names contained in this work which have been issued with identifiers by IPNI will eventually be made available to the Global Names Index. The IPNI can be accessed and the associated information contained in this publication viewed through any standard web browser using the web address http://ipni.org/. The online version of this work is archived and available from the following digital repositories: PeerJ, PubMed Central, and CLOCKSS.

## Material collection

The new species has been monitored in the field and nursery of The Gesneriad Conservation Center of China (GCCC) by the authors since the plants were collected. We collected leaf materials of this possible new species, using silica gel to dry them in the field for DNA extraction.

## Morphological observations and specimens examined

A study of the genus *Primulina* from S and SW China and adjacent areas of Vietnam was undertaken. All available specimens of *Primulina* stored in the following herbaria from China, Vietnam, United States and the United Kingdom were examined: E, GH, HN, IBK, IBSC, K, MO, KUN, PE, US and VMN (herbarium acronyms according to Index Herbariorum; *Thiers, 2017*). All morphological characters were studied using a dissecting microscope (SZX16, Olympus, Tokyo, Japan). Characters were described using the terminology presented by *Wang et al. (1998)* were applicable. The morphological comparison with other species was based on study of live plants in the field and in cultivation in GCCC, herbarium specimens, and also information gathered in the literature searches.

## DNA extraction and PCR

The DNA was extracted using a DNeasy plant mini kit (Qiagen, Crawley, UK) following the manufacturer's protocol. The extractions were checked for quality and quantity using a RapidHIT200 (IntegenX, Pleasanton, CA, USA). Total genomic DNA was used as the template for polymerase chain reactions (PCR). The PCR conditions and cycle sequencing reactions followed *Möller et al. (2009)* and *Möller et al. (2011)*.

## Phylogenetic analyses

For the phylogenetic analyses, sequences were newly acquired for each population of this new species and also the morphologically similar species *P. bobaiensis* Li, Pan and Zhang (Table 1), were added to a reduced Old World Gesneriaceae matrix of *Middleton et al. (2014)* and *Middleton et al. (2015)* and the matrices realigned manually. The resulting matrix contained 185 samples (177 species) with 26 species of *Primulina* included, covering all 39 genera currently recognized in the advanced Asiatic and Malesian Gesneriaceae at present (*Möller et al., 2009*; *Möller et al., 2011*; *Weber, Clark & Möller, 2013*; *Middleton et al., 2014*; *Middleton et al., 2015*). The tree was rooted with the outgroup taxon *Tetraphyllum* (*Möller et al., 2009*). Data for the phylogenetic analyses were downloaded from GenBank (https://www.ncbi.nlm.nih.gov/genbank/). Species, voucher information, and NCBI accession numbers are listed in Appendix S1.

Combinability of the ITS and *trn* LF datasets were investigated by the incongruence length difference (ILD) test (*Farris et al., 1995a*; *Farris et al., 1995b*), implemented in PAUP* 4.0a146 (*Swofford, 2002*) as a partition-homogeneity test (PHT). Phylogenetic analyses were performed using Bayesian inference (BI) following *Möller et al. (2011)* and *Weber et al. (2011a)*. The model GTR + I + G was selected as the optimal model for both DNA regions based on the the Akaike Information Criterion (AIC; *Akaike, 1974* in MrModeltest version 2.3 *Nylander, 2004*). Bayesian inference analyses were carried

**Table 1 Diagnostic character differences between *Primulina anisocymosa* and its close relative *P. bobaiensis*.**

| characters | *P. anisocymosa* | *P. bobaiensis* |
|---|---|---|
| Petiole | Sessile or subsessile | Petiolate, 3–12 mm long |
| Leaf blade shape | Oblong-rhombic, or elliptic-oblong; base cuneate-attenuate, and decurrent into slightly broad wings of extremely inconspicuous petiole | Elliptic or oval; base cuneate to broadly cuneate |
| Leaf blade margin | Obviously regularly 6–8 triangle to obtuse-triangle serrate from the base | Shallowly 12–14 serrate |
| Indumentum of leaf blade | Densely pubescent and sparsely villous on both surfaces | Pubescent on both surfaces |
| Peduncle indumentum | Pubescent sparsely glandular-pubescent | Pubescent |
| Bract | 1–2(–3), usually withered when flowering, free, narrowly triangular; 2–3× ca. 1 mm; outside puberulent, inside verrucose and glabrous | Opposite, oblanceolate; 8–22× 4–9 mm; pubescent on both sides |
| Corolla | Outside glandular pubescent, inside glabrous | Outside pubescent, inside sparsely puberulent |
| Calyx lobes size | 5–8× 4–5 mm | 3–5× 0.8–1.1 mm |
| Filaments indumentum | Glandular pubescent | Glabrous |
| Staminodes | 2 | 3 |

out in MrBayes version 3.2.2 (*Ronquist & Huelsenbeck, 2003*). Two independent analyses of 10,000,000 generations each were run with four Markov chain Monte Carlo (MCMC) chains and a tree sampled at every 1,000th generation. The first 10 percent of the generations were discarded as burn-in and Bayesian Posterior probabilities (BPP) obtained from the analysis were used to indicate the support for various branches.

## Chromosome preparations

Seeds for chromosome analysis were collected from the two wild populations in 2012. Voucher specimens (vouchered as: *Fang Wen 201090406-1, SLY160515-01*) have been deposited in IBK. Root tips were collected from leaf-cuttings started in vermiculite in the culture room at GCCC. Usable roots were gathered and pretreated in 2 mM 8-hydroxyquinoline at 15–18 °C for about 6 h, then fixed overnight in an ethanol-acetic acid solution (3:1) at 4 °C. The root tips were macerated with 1 N hydrochloric acid (10:1). The root tips were then stained and squashed in 2% acetic orcein. Metaphase plates were photographed using an Olympus BX51 microscope with Olympus DP71 camera attachment (Olympus, Tokyo, Japan). The chromosome numbers were determined in at least 20 cells with well-spread chromosomes of 10 different root tips from the two known populations.

## RESULTS

### Molecular phylogenetic studies

The data matrix was assembled and included 1,381 characters, including 608 for ITS and 773 for *trn* L-F. The results did not indicate incongruent phylogenetic signals ($P = 0.36$) and the two matrices were analysed together. The phylogenetic tree (Fig. 1) was highly resolved with a topology consistent with previous phylogenetic analyses by *Middleton et al. (2014)* and *Middleton et al. (2015)*. The new species was nested in a clade within *Primulina*

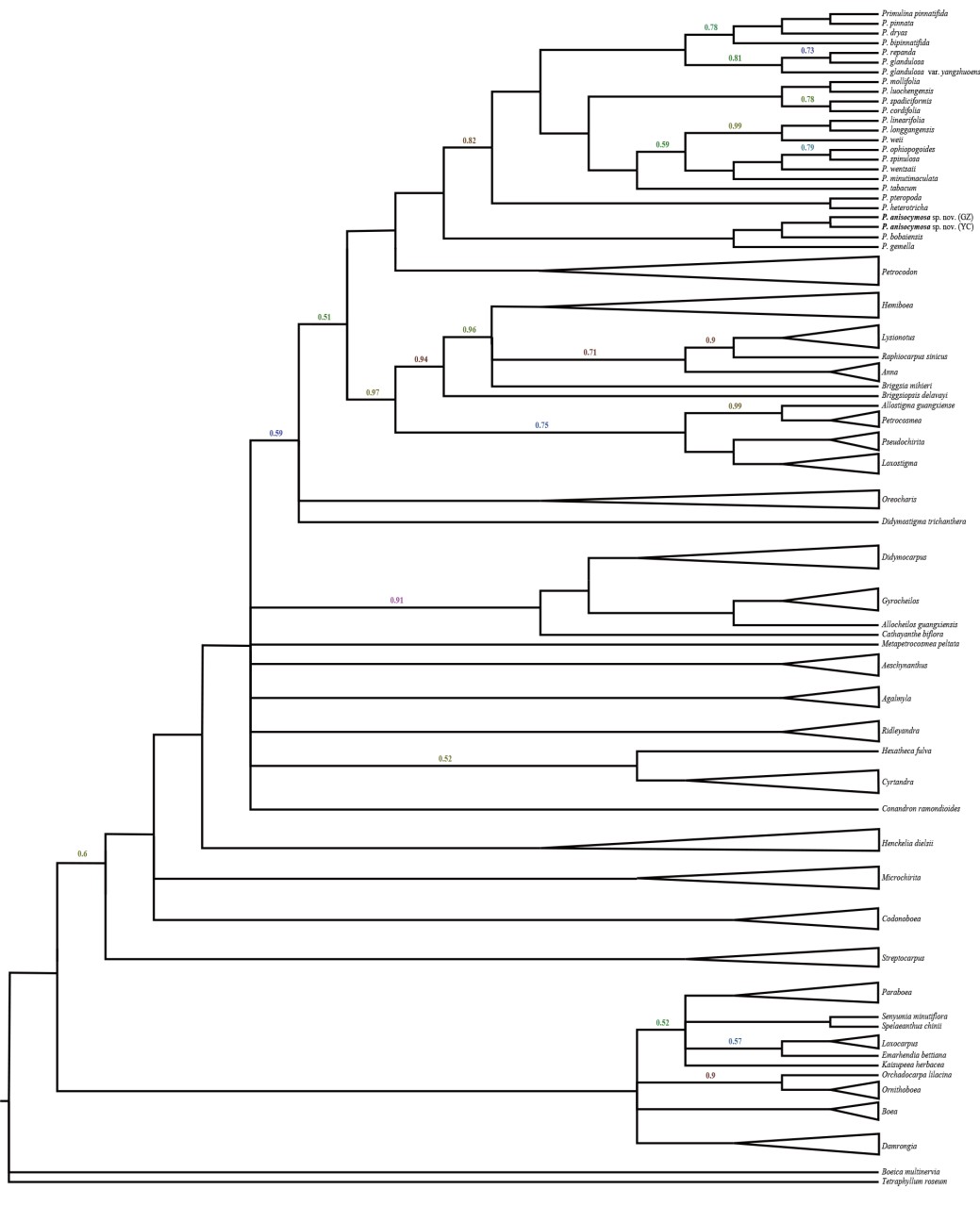

**Figure 1 Molecular result showing two populations of this new species was strongly supported in a clade, and fell deeply nested in a monophyletic *Primulina*.** Bayesian inference tree focusing on *Primulina* inferred from combined *trnL-F* + ITS data, Bayesian posterior probabilities values less than 1.00 are labeled along branches, all other nodes had PP values of 1.00. (**GZ**: type locality, Gaozhou city; **YC**: Yangchun city).

with high branch support (PP = 1.00). Two samples from different populations identified as *Primulina anisocymosa* are monophyletic (PP = 1.00). The new species was strongly supported in a clade (PP = 1.00) comprised of *Primulina gemella* (D Wood) YZ Wang and its morphologically most similar congener *P. bobaiensis*.

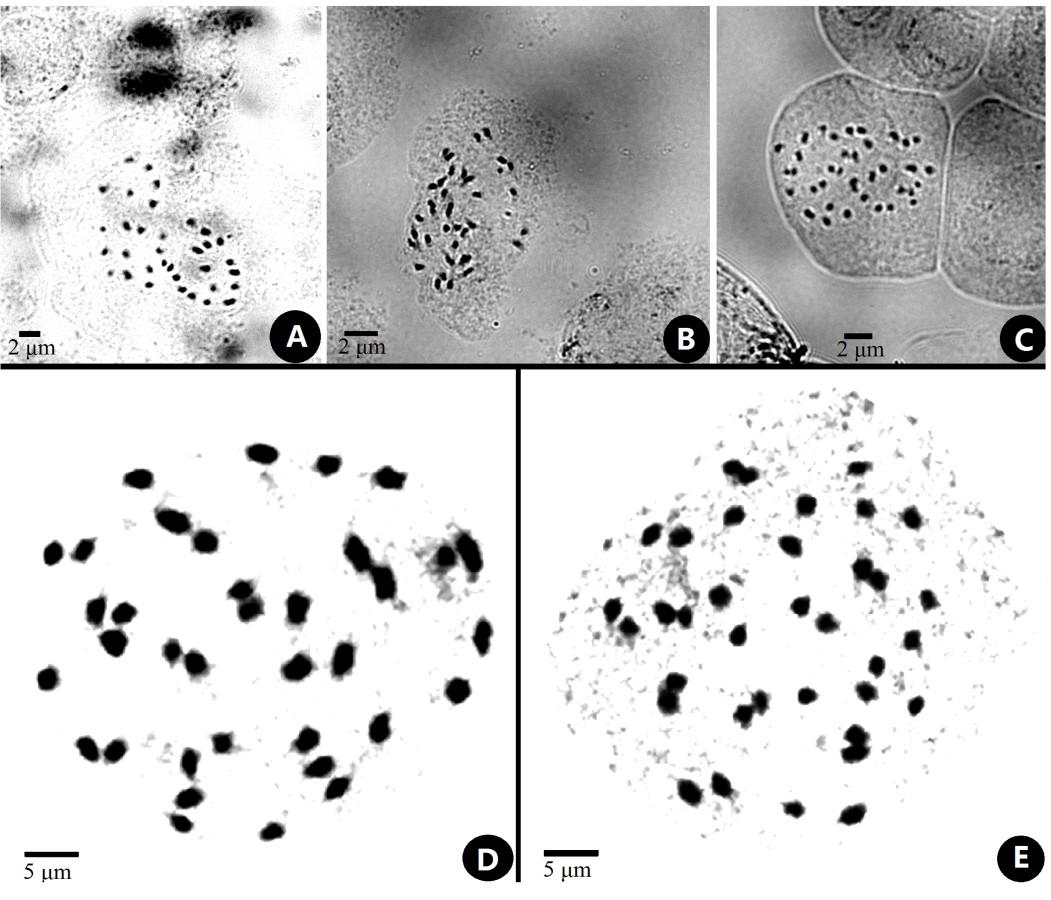

**Figure 2** **Somatic chromosomes at metaphase of *Primulina anisocymosa*.** Somatic chromosomes at metaphase of *Primulina anisocymosa* (2n = 36, from the leaf cuttings). (A–C). Microphotograph from different cells (A, B: Gangzhou, E: Yangchun). D, E: Photomicrographs of mitotic metaphase chromosomes (D: Gangzhou, E: Yangchun). Photos by Ms. Sun-Lan Yin.

## Chromosome characteristics

The chromosome size of *Primulina* was measured and noted as small with a range from 0.6–2.4 μm (*Lima-de faria, 1980*; *Yang et al., 2012*). The somatic chromosomes at metaphase of *P. anisocymosa* were photographed (Fig. 2) and determined to be diploid with 2n = 36.

## Taxonomic treatment

*Primulina anisocymosa* F. Wen, Xin Hong & Z.J. Qiu, sp. nov. (Figs. 3 and 4).

## IPNI

**Type.** China. Guangdong: Gaozhou city, rocky crevices in moist shady cliffs on a red sedimentary rock hill, elevation ca. 120 m, 20 Nov. 2012, *F. Wen 20121120* (holotype IBK!; isotype ANU!).

**Additional collections.** China. Guangdong: Yangchun, ca. 107 m, 06 April 2009, *Fang Wen 201090406-1* (IBK!; ANU!); China. Guangxi, cultivated in nursery of GCCC, Guilin

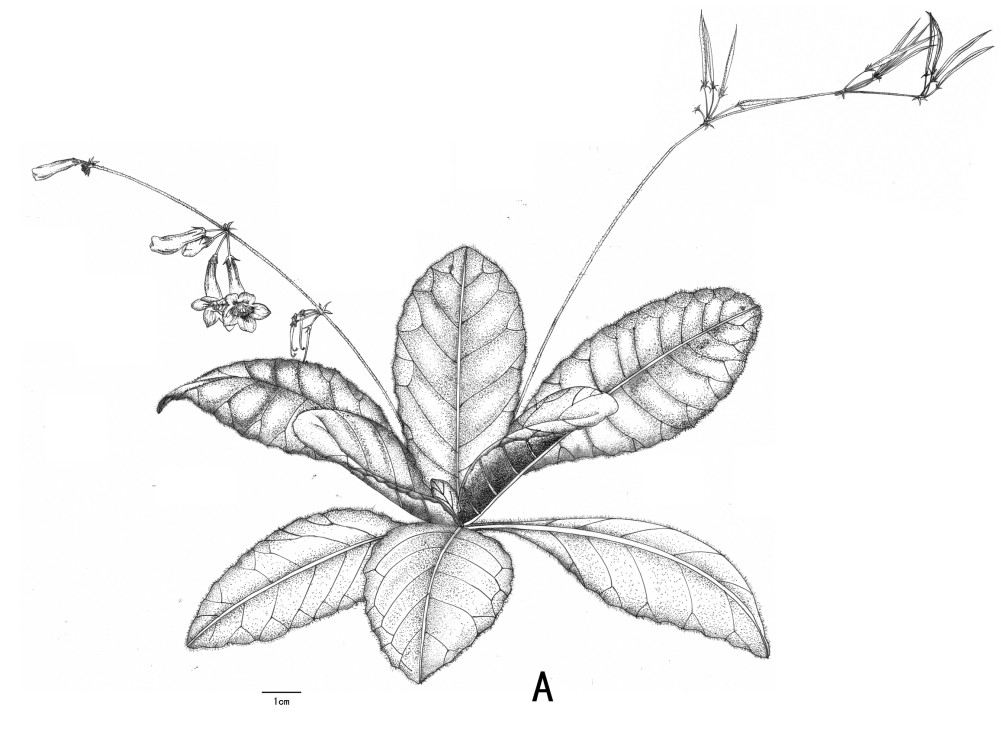

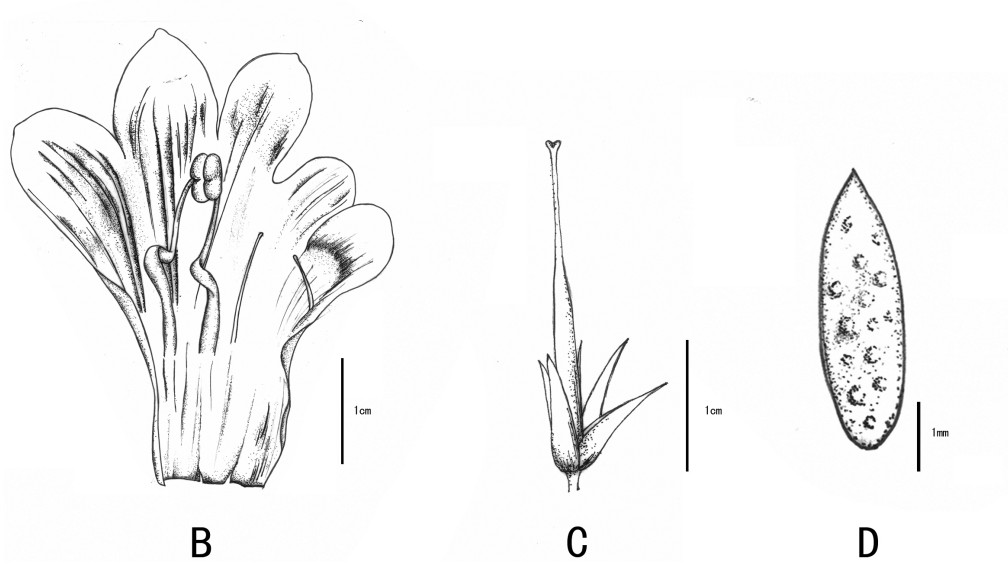

**Figure 3** **Line-drawing of *Primulina anisocymosa*.** *Primulina anisocymosa* F Wen, Xin Hong and ZJ Qiu. (A) Plant in flower. (B) Corolla opened showing stamens and staminodes. (C) Calyx and pistil with ovary, style and stigma. (D) Seed. —Drawn by Ms. Xiao-Ming Xu and Ms. Wen Ma.

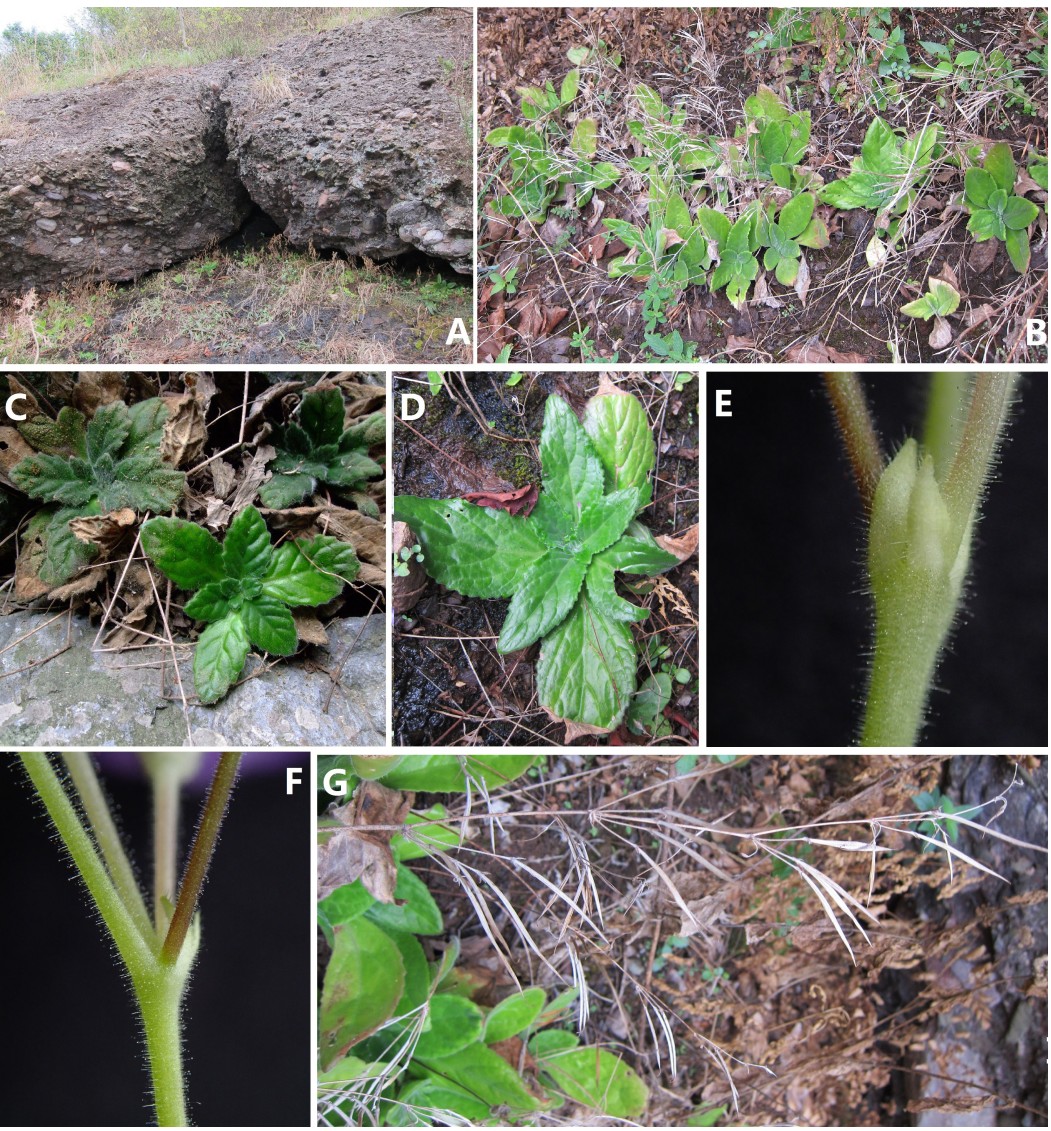

**Figure 4 Photos of *Primulina anisocymosa* two populations in natural habitat.** *Primulina anisocymosa* (F Wen, Xin Hong and ZJ Qiu) (A, B) Habitat (A, Gaozhou; B, Yangchun) (C, D) Vegetative part of plants (C, Yangchun; D, Gaozhou) (E) Bracteoles, showing not-paired, aligned on one side at the base of pedicel. (F) Cymule, reduced to the point attachment and forming swollen nodules at the base. (G) Zigzag monochasial infructescence. Photos by Fang Wen.

Botanical Garden, introduced from Gaozhou city, growing on rocky crevices in moist shady cliffs on a red sedimentary rock hill, elevation ca. 120 m, *SLY160515-01* (IBK!).

**Diagnosis.** *Primulina anisocymosa* differs from other congeners by the presence of a zigzag monochasial cyme.

**Description.** Perennial herb, acaulescent. Rhizome subterete, 1–3 cm long, 1.5–2 cm diam, glabrous. Leaves 6–9 or more, 3-whorled, basal; sessile or subsessile. Leaf blade slightly fleshy, dried papery, oblong-rhombic, or elliptic-oblong, 3–9(–12) ×2.5–6(–8)

cm, apex obtuse or sometimes rounded, base cuneate to attenuate, and decurrent into slightly broad wings of extremely inconspicuous petiole, or nearly sessile, margin distinctly serrate with 6–8 triangular serration on each side; densely pubescent intermixed with sparsely villous trichomes on both surfaces, lateral veins 4–7 on each side, impressed adaxially and prominent abaxially. Cyme monochasial, reduced with 4–6 (–10) or more cymules alternating along rachis (appearing zigzag), 3–6 flowers per cymule. Peduncles 15–20 cm long, ca. 2 mm diam, pubescent intermixed sparsely with glandular trichomes. The inflorescence has only one of the two lateral paraclades of the pair-flowered cyme extended. The others are reduced to the point of pedicel attachment and form the swollen nodules at the base of 2–3(–5) flowers. All flowers of each cymule clustered and each cluster facing away from the rachis. Bracts variable, usually 1–2, occasionally 3, sometimes withered when flowering, free, narrowly triangular, 2–3× ca. 1 mm, outside puberulent, inside verrucose and glabrous, margin entire, apex acute, aligned on one side of the rachis; bracteoles three, subulate, 1× ca. 0.2 mm, not-paired, aligned on one side at the base of pedicel. Pedicels extended only on one side, 18–20 mm long, ca. 1.0 mm in diameter, densely glandular-puberulent. Calyx 5 lobed from base; segments equal, lanceolate-oblong, 5–8× 4–5 mm, outside densely puberulent, inside verrucose and glabrous, margin entire, apex acute. Corolla ca. 1.5 cm long, purpe, upper part of the interior of the corolla with two dark brown short stripes; mouth 0.6–0.8 cm diam, outside glandular-pubescent, inside glabrous; tube nearly infundibuliform, ca. 1.3 cm long, limb distinctly 2-lipped, adaxial lip 2-parted for over half its length, lobes obliquely ovate, ca. 0.5 cm long, 0.3–0.4 cm in diameter at base, abaxial lip 3-parted to the base, central lobe ovate oblong, 0.5–0.6 cm long, 0.25–0.35 cm in diameter, lateral ones obliquely oblong. Stamens 2, adnate to ca. 6 mm above the corolla base; filaments ca. 6 mm long, glandular-pubescent, geniculate at the base, anthers fused by their entire adaxial surfaces, elliptic, ca. 2 mm long, yellowish brown, glabrous; staminodes two, linear, ca. 3 mm long, glabrous, adnate to ca. 7 mm above the corolla base, capitate at apex. Disc annular, ca. 1 mm high, margin entire. Ovary linear, 0.9–1.1 cm long, ca. 2 mm in diameter, densely glandular- and eglandular-puberulent; style green, ca. 6 mm long, ca. 0.8 mm in diameter, glandular-puberulent. Stigma translucent to green, obtrapeziform, apex retuse, ca. 1.2 mm long. Capsule linear, 4–4.5 cm long, ca. 2 mm in diameter, glandular- and eglandular-puberulent. Seeds long-ellipsoidal, dark brown, mammillate on the surface.

**Distribution.** The new species is known from two locations: Gaozhou City and Yangchun City, Guangdong Province, southern China. This area is in the transitional zone between the tropics and subtropics.

**Habitat and ecology.** *Primulina anisocymosa* is locally abundant in Yangchun, although very rare in Gaozhou, Guangdong. The total number of this species in Gaozhou does not exceed 100 individuals. It grows in rocky crevices of moist shady cliffs on a red sedimentary rock hill, at an elevation of 50 m a.s.l in Gaozhou City. The species also grows on the moist rock surface of limestone cave entrances in Yangchun. The average annual temperature of the two localities is similar (ca. 21 °C) and the average annual precipitation is around 2,380 mm. *P. anisocymosa* occurs in subtropical evergreen broad-leaved forest. Flowering from September to October and fruiting from October to December.

**Etymology.** The scientific name is derived from its unusual zigzag monochasial cyme. The latin prefix, "*aniso-*", means different, uneven or asymmetrical; "*cyma*" refers to the predominant inflorescence type seen in *Primulina* a pair-flowered cyme.

## DISCUSSION

The size of *P. anisocymosa* chromosomes is within the above-mentioned range for the genus. At present, almost all known chromosome numbers in this genus appear to represent a diploid number with a basic number of $x = 18$, except the polyploid *P. longgangensis* (Wang) Liu and Wang with $2n = 72$ (*Christie, Barber & Möller, 2012*; *Möller et al., 2002* onwards). Our chromosome counts of ($2n = 36$) for *P. anisocymosa* corresponds with known chromosome number ($x = 18$) supporting its generic placement in *Primulina*.

The pair-flowered cyme is the basic type of inflorescence in the Old World Gesneriaceae (*Weber, 1978*; *Weber, 1982*). The inflorescences in *Primulina* are usually dichotomous cymes with few to many flowers or simplified single-flowered cymes (*Li & Wang, 2004*; *Wang et al., 1998*; *Chen et al., 2008*; *Wei et al., 2010*). The development of pair-flowered cymes shows that the main axis of the inflorescence has a true terminal flower and a front flower. Two lateral paraclades arise in the axils of two opposite lateral bracts below the terminal flower, the following branches continuously repeat the same pattern (*Weber, 1978*; *Weber, 1982*; *Weber, 2013*; *Wang & Li, 2002*; *Li & Wang, 2004*). This new species has an inflorescence morphology unlike any other described *Primulina* species to date.

The branching morphology of the inflorescences of this new species is a zigzag monochasial branching structure. The inflorescence has only one of the two lateral paraclades of the pair-flowered cyme extended. The others are reduced to the point of pedicel attachment and form the swollen nodules at the base of 2–3(–5) flowers. Each flower cluster whorl turns in the direction of the main axis and flowering continues upward in an acropetal sequence. Frthermore, the markedly special character of inflorescence in two known populations is very stable, as long as the plants grow normally. This is morphologically distinctive from all known *Primulina* species.

The zigzag monochasial inflorescence and infructescence of *Primulina anisocymosa* is so peculiar that we can easily distinguish it from other known species in the genus. It morphologically resembles *P. bobaiensis* (see Fig. 5) in having similar corolla shape and color. Both species are found in southern China (Guangdong and Guangxi), but *Primulina anisocymosa* can be easily distinguished by the inflorescence architecture.

## ACKNOWLEDGEMENTS

The authors are grateful to Ms. Xiao-Ming Xu and Ms. Wen Ma for drawing the handsome illustration. The authors also should like to thank Mr. Wei-Jun Wu for his advice on the living specimen collection, and Ms. Lan-Ying Su for her wonderful Chromosome cytology works.

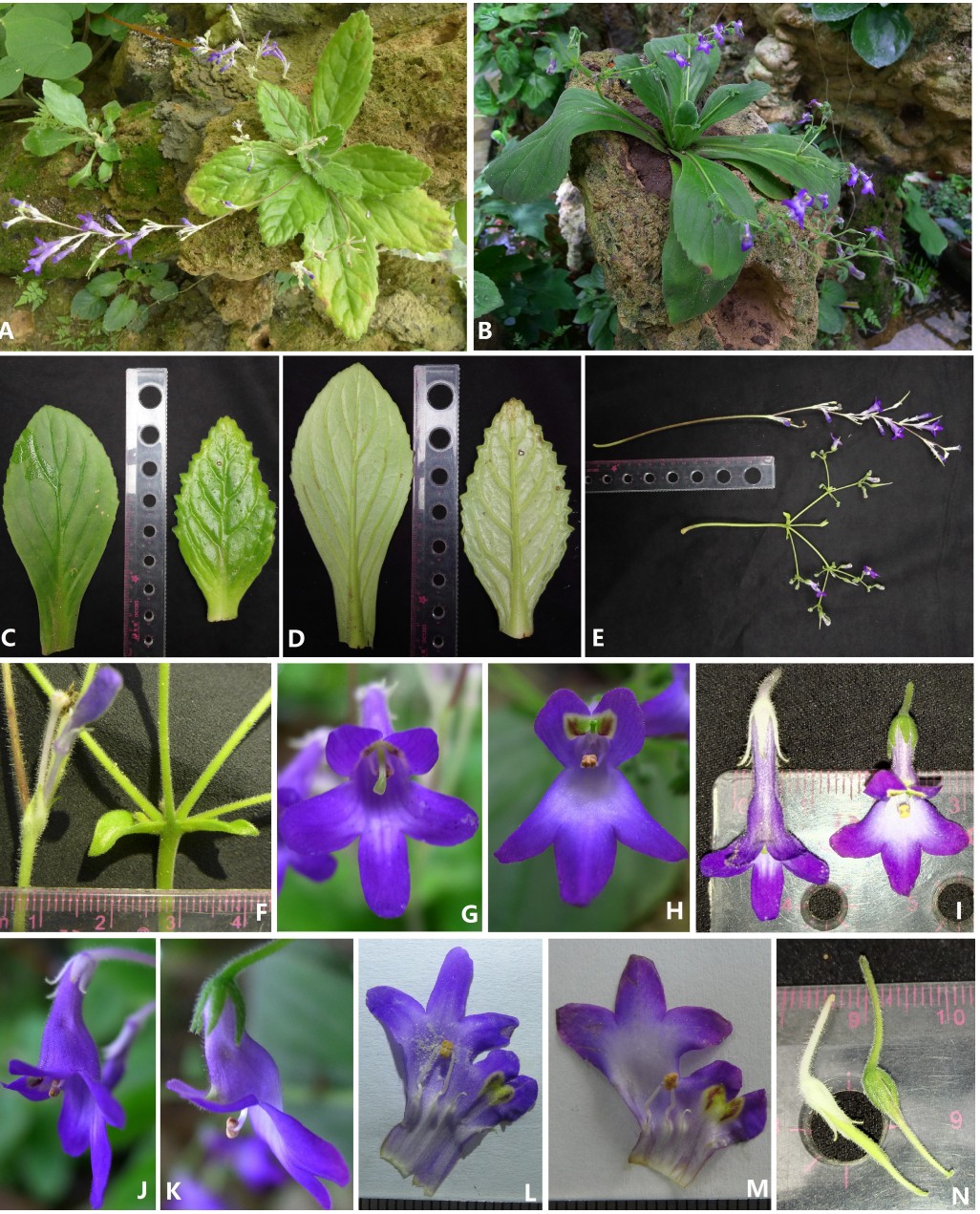

**Figure 5  Comparison of *Primulina anisocymosa* and *P. bobaiensis*.** Comparison of *Primulina anisocymosa* and *P. bobaiensis*, (A) Habit of *P. anisocymosa* when in flower. (B) A. Habit of *P. bobaiensis* when in flower. (C) Adaxial leaf blades (left: *P. bobaiensis*, right: *P. anisocymosa*). (D) Abaxial leaf blades (left: *P. bobaiensis*, right: *P. anisocymosa*). (E) Cymes (upper: *P. anisocymosa*, lower: *P. bobaiensis*). (F) Bracts (left: *P. anisocymosa*, right: *P. bobaiensis*). (G) Frontal view of corolla of *P. anisocymosa*. (H) Frontal view of corolla of *P. bobaiensis*. (I) Top view of corolla (left: *P. anisocymosa*, right: *P. bobaiensis*). (J) Lateral view of corolla of *P. anisocymosa*. (K) Lateral view of corolla of *P. bobaiensis*. (L) Opened corolla of *P. anisocymosa*. (M) Opened corolla of *P. bobaiensis*. (N) Pistils without corolla (left: *P. anisocymosa*, right: *P. bobaiensis*). Photoed by Xin Hong and Fang Wen.

### Funding

This work was supported by the Anhui Province Resources Information Center Development Project, the Anhui University Doctor Startup Fund, the Undergraduate Innovation and Entrepreneurship Training Program, the Key University Science Research Project of Anhui Province (KJ2017A022), the Fund of Guangxi Key Laboratory of Plant Conservation and Restoration Ecology in Karst Terrain (No.17-259-23), the Nellie D. Sleeth Scholarship Endowment Fund by Gesneriads Society, the Boyce Edens Research Fund by the African Violet Society of America, the Guangxi Natural Science Foundation (2015GXNSFBB139004) and the Key Research and Development Project of Guangxi (Guike AB16380053). The funders had no role in study design, data collection and analysis, decision to publish, or preparation of the manuscript.

### Grant Disclosures

The following grant information was disclosed by the authors:
Anhui Province Resources Information Center Development Project.
Anhui University Doctor Startup Fund.
Undergraduate Innovation and Entrepreneurship Training Program.
Key University Science Research Project of Anhui Province: KJ2017A022.
Fund of Guangxi Key Laboratory of Plant Conservation and Restoration Ecology in Karst Terrain: 17-259-23.
Nellie D. Sleeth Scholarship Endowment Fund by Gesneriads Society.
Boyce Edens Research Fund by African Violet Society of America.
Guangxi Natural Science Foundation: 2015GXNSFBB139004.
Key Research and Development Project of Guangxi: Guike AB16380053.

### Competing Interests

The authors declare there are no competing interests.

### Author Contributions

- Xin Hong and Fang Wen conceived and designed the experiments, performed the experiments, contributed reagents/materials/analysis tools, prepared figures and/or tables, authored or reviewed drafts of the paper, approved the final draft.
- Jeremy Keene conceived and designed the experiments, analyzed the data, authored or reviewed drafts of the paper, approved the final draft.
- Zhi-Jing Qiu conceived and designed the experiments, contributed reagents/materials/-analysis tools, approved the final draft.

### Data Availability

The group sequences described here are accessible via GenBank accession numbers MK131751 to MK131756, and are available as Supplemental Files.

## New Species Registration

The following information was supplied regarding the registration of a newly described species:

  *Primulina anisocymosa* F. Wen, Xin Hong & Z.J. Qiu, sp. nov. LSID 77192338-1.

## Supplemental Information

Supplemental information for this article can be found online at http://dx.doi.org/10.7717/peerj.6157#supplemental-information.

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
