# Peer review of "Primulina anisocymosa (Gesneriaceae), a new species with a unique inflorescence structure from Guangdong, China"

_PeerJ, doi:10.7717/peerj.6157_

## Round 0.1 · original submission · Minor Revisions

This is a good paper, just in need a bit of tweaking (minor revision).

·

Basic reporting

No critical comments.
The manuscript is well-written. Language usage is clear, with only a few points requiring clarification which are detailed below.
Literature cited is appropriate to the topic and the introduction provides suitable context.
The article is suitably structured and novel data are made available in appendices.
Results and discussion are suitably correlated.

Experimental design

No critical comments.
The manuscript is well conceived with respect to methodology, analyses and conclusions.
The paper sets a high standard for the recognition of a new species.

Validity of the findings

No critical comments.
The conclusions of the paper are sound and the recognition of a new species is well-supported.

Additional comments

The following minor points are noted for consideration or correction:

Line 27: ‘generic’ is not needed here as it is followed by ‘in the genus’
Line 59 and elsewhere: Check the journal style as to whether initials in citation are appropriate, or if repeated dates with the same surnamed should be lettered to distinguish different authors.
Lines 72–80: This information is generally assumed to be true, so would usually be left out, but it may be in keeping with journal standards to include these statements?
Line 84: Change ‘these’ to ‘this’
Line 103: Change: ‘the newly acquired sequences of’ to: ‘sequences were newly acquired for’
Line 10: Change: ‘reduce’ to ‘reduced’
Line 113: ‘trn’ should be in italics.
Lines 167 and 168: Change ‘triangle’ to ‘triangular’
Lines 172–173: It would be good to clarify the position of the vestigial flowers on the inflorescence further. Lines 228–229 present this more clearly.
Line 175: You vary whether or not a space is used to separate bracketed measurements through the description – please check the recommended journal style.
Line 179: I suggest changing ‘sect’ to ‘partite’ or ‘merous’
Lines 201–202: change ‘caves entrance’ to ‘cave entrances’
Line 202: Add: ‘the’ before ‘two localities’
Line 215: Add: ‘for’ before ‘P. anisocymosa’
Line 217: Delete ‘the’ before ‘inflorescence’
Line 231: Change: ‘that of other’ to all’
Line 334: ‘Petrocodon’ should be in italics.
Line 338: The closing bracket is missing.

Figure 1 caption
Line 1: Change ‘shoeing’ to ‘showing’; delete second ‘i' in ‘populatiions’

Figure 2 caption
Line 1: Change ‘Hand-‘ to ‘Line ‘
Note that the angle of presentation of the inflorescence makes it difficult to determine the extent of the ‘zig-zag’ arrangement that is diagnostic for the species, so this may be best updated.

Figure 3 caption
Line 2: ‘sp. nov.’ is not required here – otherwise also add to Fig. 2 caption.
Line 6: Change ‘Photoed’ to ‘Photos’ or ‘Photographs’

Figure 4 caption
Line 2: Change ‘Habitat in flowering’ to ‘Habit when in flower’. As A2 in photographed in a cultivated environment, it is not suitable to describe this as ‘habitat’.

Table 1
Text flow would be clearer if all aligned to the top of the cells.

Reviewer 2 ·

Basic reporting

This is a well written and very comprehensive piece of research. It is well referenced and the authors are aware of the relevant literature in the field.

The legends for the figures are not as well written. The first part of the legend for Figure 1, for example, is not necessary as the results are discussed in the main text and more information about what type of tree it is would be useful eg maximum clade credibility tree.

Experimental design

This is a comprehensive study allowing the authors to confidently describe this as a new species and place it in the correct genus.

Validity of the findings

Although placement of this new species in Primulina is well argued, a comment is made in the introductory section about the highly variable nature of the inflorescence, even within an individual and, yet, it is an inflorescence character that is selected to differentiate this species from all others in the genus. Although I feel confident that the authors are correct, and the character is striking, I think it would be worth adding a statement about the apparent lack of variability in this character in the specimens seen of this new species.

Annotated reviews are not available for download in order to protect the identity of reviewers who chose to remain anonymous.

---

## Round 0.2 · Minor Revisions

Please, see the attached file for the last small edits to be performed.

---

## Round 0.3 · accepted · Accept

You paper is now in shape to be accepted. Congratulations!

#